# Every Subtlety Counts: Fine-grained Person Independence Micro-Action Recognition via Distributionally Robust Optimization

## Abstract

Micro-action Recognition (MAR) is vital for psychological assessment and human-computer interaction. However, existing methods often fail in real-world scenarios due to inter-person variability, *e.g.*, differences in motion styles, execution speed, and physiques, cause the same action to manifest differently, hindering robust generalization. To overcome this, we propose the ***Person Independence Universal Micro-action Recognition Framework*** (*PIUmr*), which embeds Distributionally Robust Optimization (DRO) principles to learn person-agnostic representations. *PIUmr* achieves this through two synergistic, plug-and-play components that operate at the feature and loss levels, respectively. First, at the feature level, the ***Temporal–Frequency Alignment Module*** (*TFAM*) normalizes person-specific motion characteristics. It employs a dual-branch architecture to disentangle motion patterns. The temporal branch uses Wasserstein-regularized alignment to create a stable dynamic trajectory, mitigating variations caused by different motion styles and speeds. The frequency branch uses variance-guided perturbations to build robustness against person-specific spectral signatures arising from different physical attributes (*e.g.*, skeleton size). A consistency-driven mechanism then adaptively fuses these branches. Second, at the loss level, the ***Group-Invariant Regularized Loss*** (*GIRL*) is applied to the aligned features to guide robust learning. It simulates challenging, unseen person-specific distributions by partitioning samples into pseudo-groups. By up-weighting hard boundary cases and regularizing subgroup variance, it forces the model to generalize beyond easy or frequent samples, thus enhancing its robustness against the most difficult person-specific variations. Extensive experiments on the large-scale MA-52 dataset demonstrate that *PIUmr* significantly outperforms existing methods in both accuracy and robustness, achieving stable generalization under fine-grained conditions.

## 1 Introduction

Micro-action recognition (MAR) focuses on modeling short-lived, low-amplitude human movements that often occur unconsciously (Guo et al., 2024), such as brief eye twitches, subtle posture adjustments, or fine-grained micro-gestures. These subtle behaviors are closely tied to latent cognitive or affective states (Chen et al., 2023; Wang et al., 2024) and thus play an essential role in applications ranging from psychological assessment and deception detection to human–machine empathy (Allaert et al., 2022; Lu et al., 2025). Compared with conventional action recognition tasks (Kay et al., 2017), MAR is considerably more challenging due to the subtle movements during the short time period. Its signals are inherently weak, easily obscured by contextual noise, and exhibit high inter-class similarity alongside pronounced intra-class variability (Gu et al., 2025a), all of which amplify its susceptibility to distributional shifts.

A particularly critical source of distributional shift in MAR is inter-person variability. In real-world settings, the same micro-action may manifest with different temporal rhythms, spectral patterns, or intensity levels depending on individual motion styles, execution speed, and skeleton size. Such heterogeneity causes unstable feature representations and poor generalization in cross-person evaluation, making person independence a fundamental yet underexplored challenge for MAR.

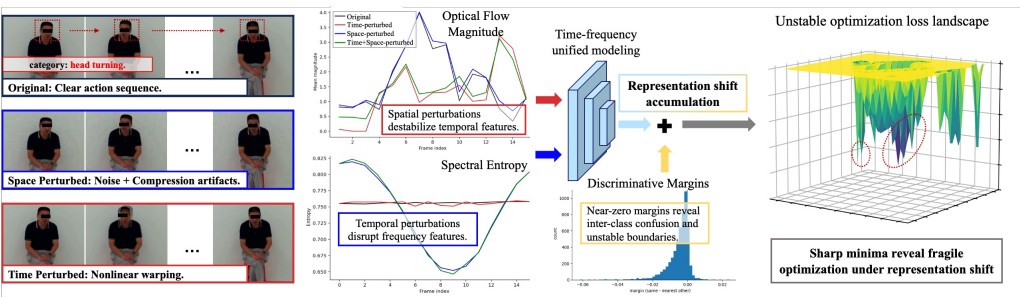

Figure 1: Illustration of representation shift in micro-action recognition. Spatial perturbations desta-bilize temporal features, and temporal perturbations disrupt frequency features. Together with mar-gin collapse that reflects inter-class confusion and unstable boundaries, these shifts give rise to sharp minima in the loss landscape, ultimately leading to fragile optimization.

Although recent advances in spatiotemporal modeling and fine-grained supervision (Lin et al., 2019; Feichtenhofer et al., 2019) have improved recognition accuracy, these methods are primarily based on empirical architectural designs or heuristic training tricks, without principled mechanisms to ensure robustness across individuals. Consequently, these models inevitably overfit to dominant persons, fail to reconcile heterogeneous cues across people, and generalize poorly to ambiguous or boundary cases (Li et al., 2025; Gu et al., 2025a).

As illustrated in Fig. 1, we argue that these limitations can be systematically understood under the lens of distributional shift in person-independent MAR, which manifests in three interrelated forms: **1) Temporal–frequency inconsistency.** Temporal representations are easily perturbed by frame jitter, rhythm variations, and occlusions, while frequency representations, though more structural, are highly sensitive to person-specific factors such as motion amplitude and style. This asymmetric vulnerability destabilizes time–frequency modeling across different individuals. **2) Cross-modal imbalance under inter-person perturbations.** Person variability often induces uneven drifts be-tween temporal and frequency pathways. Static or heuristic fusion cannot adaptively correct such an imbalance, and may even amplify unreliable modalities, thereby weakening person-independent discrimination. **3) Subgroup vulnerability.** Owing to their transient and subtle nature, many micro-actions appear near inter-class decision boundaries, especially for minority or person-specific sub-groups. Pioneer optimization tends to emphasize frequent and easy instances, while neglecting rare or ambiguous ones, resulting in fragile boundaries and reduced robustness in cross-person scenarios.

To address this fundamental lack of robustness against inter-person variability, we introduce the ***Person Independence Universal Micro-action Recognition Framework*** (*PIUmr*). Our framework systematically embeds principles from Distributionally Robust Optimization (DRO) to learn person-agnostic representations that remain stable even in worst-case scenarios dominated by heteroge-neous motion styles. *PIUmr* achieves this through two synergistic modules. Firstly, the ***Tempo-ral–Frequency Alignment Module*** (*TFAM*) normalizes feature representations to make them invari-ant to individual motion signatures. Its dual-branch architecture explicitly disentangles and aligns motion characteristics. The temporal branch uses Wasserstein-regularized alignment to stabilize dynamic trajectories, mitigating variations caused by different execution styles and speeds. The frequency branch employs variance-guided perturbations to counteract person-specific spectral dis-tortions arising from different physiques, *e.g.*, different skeleton sizes. These branches are then adaptively integrated via a consistency-driven fusion mechanism that prioritizes the more robust representation. Secondly, the ***Group-Invariant Regularized Loss*** (*GIRL*) guides the training pro-cess to generalize across diverse, unseen individuals. It simulates latent person-specific distributions by creating pseudo-groups and then up-weights challenging boundary samples. By regularizing sub-group variance, *GIRL* prevents the model from overfitting to common motion patterns and ensures it learns effectively from rare or atypical examples, thereby enhancing its worst-case robustness. Totally, our main contributions are summarized as follows:

- As best of our knowledge, we are the first to introduce the principle of DRO into MAR explicitly from the perspective of person independence. By unifying temporal inconsistency, cross-modal imbalance, and subgroup vulnerability as manifestations of inter-person distributional variabil-ity, we provide a principled explanation of MAR's core bottlenecks and formulate a systematic solution grounded in robustness theory.

- To create feature representations that are invariant to individual motion signatures, a plug-and-play module *TFAM* is designed, which has a dual-branch architecture explicitly disentangling and normalizing person-specific characteristics. The temporal branch mitigates variations in motion style and speed by using a local–global Wasserstein regularization to produce a stable, aligned dynamic representation. The frequency branch suppresses person-specific spectral distortions, often caused by different physiques, through a novel variance-guided perturbation and adaptive activation strategy. Furthermore, a consistency-driven fusion mechanism intelligently re-weights and integrates the two branches, ensuring the final representation is robust even when an individual exhibits strong, asymmetric perturbations.

- To ensure the model generalizes robustly across all individuals, especially those with atypical motion patterns, we introduce a novel training objective *GIRL*. To improve worst-case performance, *GIRL* first partitions samples into pseudo-groups to simulate latent, person-specific data distributions. It then up-weights the most challenging and ambiguous samples on the boundaries of these groups using a Gaussian-based function, thereby forcing the model to focus its learning on "hard" cases. Finally, by regularizing the risk variance across these groups, the objective prevents the model from simply overfitting to the majority (or "easy") individuals. This strategy achieves robust, subgroup-invariant optimization and significantly improves generalization to unseen, heterogeneous populations.

- Extensive experiments on the large-scale MA-52 benchmark validate the effectiveness of our framework. *PIUmr* consistently outperforms state-of-the-art methods in both accuracy and robustness, with particularly strong gains in cross-subject evaluations and distribution-shifted conditions, confirming its ability to generalize across diverse individuals.

## 2 RELATED WORK

### 2.1 MICRO-ACTION RECOGNITION AND TEMPORAL–FREQUENCY MODELING

MAR aims to identify short-lived, low-amplitude movements that convey unconscious behavioral cues and subtle affective states (Liu et al., 2021a). Compared with conventional action recognition benchmarks such as Kinetics (Kay et al., 2017), MAR is far more challenging due to its short duration, weak motion signals, high inter-class similarity, and pronounced intra-class variability (Chen et al., 2023), which make it highly sensitive to contextual noise and person-dependent variability. A major bottleneck lies in the person-independence setting, where differences in rhythm, style, and motion scale across individuals cause unstable representations and poor cross-subject generalization. Recent benchmarks such as MA-52 (Guo et al., 2024) further expose the challenges of MAR, showing that large-scale settings are still dominated by cross-subject heterogeneity and class imbalance. These issues make it difficult for models to generalize across individuals and to handle rare or ambiguous categories. On the modeling side, approaches such as MMN (Gu et al., 2025a) incorporate motion-guided cues, while transformer-based models (Li et al., 2025) and relation reasoning strategies improve fine-grained discrimination. However, these methods often remain heuristic and lack principled mechanisms to ensure robustness across heterogeneous subjects and boundary cases.

Parallel to MAR-specific architectures, video representation learning has revealed the complementary value of temporal and frequency-domain cues. Temporal models (Wang et al., 2021; Liu et al., 2024) capture motion dynamics at multiple scales, while frequency features derived from DCT or Fourier transforms provide stability against short-term noise, compression artifacts, and person-specific appearance variations (Cui et al., 2025; Liu et al., 2021c). Although joint temporal–frequency learning, as in dual-branch DCT networks (Chen et al., 2021a) or broader spatio-temporal-frequency fusion (Chen et al., 2024), has shown potential, these methods still treat the two domains independently and lack principled alignment. Without explicitly reconciling temporal rhythms and spectral structures, models fail to robustly represent differences induced by individual variations in style, speed, or body size, leading to unstable generalization across subjects.

To bridge this gap, we argue that lacking explicit temporal–frequency integration causes unstable representations, weakening the generalization of representation learning under inter-person variability. Our *TFAM* mitigates this by disentangling temporal dynamics and frequency structures and adaptively aligning them, yielding more stable and invariant representations for person-independent MAR. Further empowered by modules *GIRL*, this forms the core of *PIUmr*.

**Person Independence Universal Micro-action Recognition Framework**

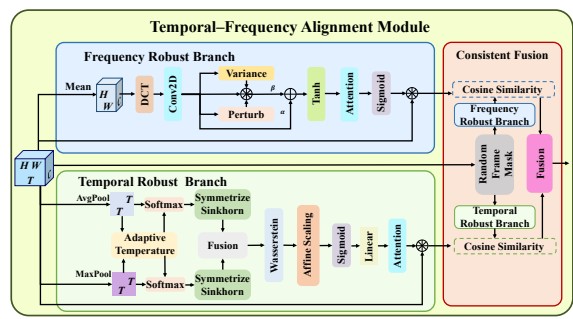

Figure 2: An overview of the proposed *PIUmr*.

## 2.2 DISTRIBUTIONALLY ROBUST OPTIMIZATION IN VISION

From the perspective of person-independent MAR systems, it is necessary to identify the samples with the largest person span and the largest difference in micro-motion features of the same category, which means the system is capable of acquiring person-invariant micro-movement features. DRO provides a principled paradigm for learning under uncertainty, where the objective is to minimize the maximum expected loss over a neighborhood of plausible distributions (Lin et al., 2022a;b). By construction, DRO focuses on worst-case risks rather than average performance, offering strong theoretical guarantees against distributional shifts. The central challenge of MAR lies in inter-person variability, where the same micro-action may exhibit diverse rhythms, spectral patterns, or intensities across individuals. From a robustness view, such variability can be regarded as hidden distributions or adversarial perturbations. This naturally aligns with DRO, which optimizes for worst-case risks over distributional neighborhoods, providing a principled basis for person-independent MAR and unifying the handling of heterogeneous styles, modality drifts, and boundary cases. On the other hand, DRO has already shown its potential in computer vision areas. For instance, the group DRO with strong regularization improves minority group accuracy while preserving overall performance (Wu et al., 2023; Sagawa et al., 2020), hardness-aware sampling incorporates DRO principles into mini-batch optimization to prioritize difficult examples (Fidon et al., 2020), and adaptive DRO-aware optimizers dynamically reweight samples to stabilize training in deep networks (Feoktistov et al., 2025). These successes also demonstrate the effectiveness of DRO in mitigating imbalance and instability. Despite these advances, DRO remains unexplored in video-based fine-grained recognition, especially MAR. Thus, to the best of our knowledge, we are the first to apply DRO in MAR, unifying temporal–frequency alignment and subgroup-invariant optimization within a coherent framework for fine-grained person-independence MAR.

## 3 METHODOLOGY

In this section, we propose *PIUmr*, a framework tailored to mitigate representation instability in MAR tasks under distributional shifts and cross-person variability. Fig. 2 depicts the overall architecture of *PIUmr*. Built upon the X3D (Feichtenhofer, 2020) backbone, it integrates two plug-and-play modules that enhance robustness at the representation and optimization levels. The *TFAM* employs a dual-branch temporal–frequency alignment with perturbation-aware fusion to stabilize cross-modal features across individuals, while the *GIRL* partitions samples into pseudo-groups and regularizes subgroup risks to achieve person-independent discrimination on hard and boundary cases.

Figure 3: The pipeline of *TFAM* in *PIUmr*.

## 3.1 TEMPORAL–FREQUENCY ALIGNMENT UNDER PERSON-INDEPENDENT ROBUSTNESS

To accurately identify the same type of micro-movements of different persons, as shown in Fig. 3, a dual-pathway architecture *TFAM* is explicitly designed to align temporal and frequency cues under distributional uncertainty. Unlike conventional empirical risk minimization, which tends to emphasize dominant modes and overlook rare or ambiguous micro-actions, *TFAM* simulates worst-case

shifts in both temporal and frequency pathways and adaptively aligns them via consistency-guided fusion. This design enhances stability and person-independent generalization across heterogeneous MAR scenarios.

### 3.1.1 FREQUENCY BRANCH: SPECTRAL ROBUSTNESS WITH PERTURBATION

By explicitly regularizing against person-specific heterogeneity in amplitude, rhythm, and style, the frequency pathway in *TFAM* aims to extract discriminative yet invariant spectral cues. Given an input sequence $x \in \mathbb{R}^{B \times T \times C \times H \times W}$, we first average across the temporal dimension and apply a bank of DCT filters $\{\mathbf{D}_d\}_{d=1}^{D}$ to generate spectral embeddings, which can be expressed as:

$$\mathbf{E} = \text{Conv2D}\Big(\tfrac{1}{T}\sum\nolimits_{t=1}^{T} x_t, \{\mathbf{D}_d\}\Big) \in \mathbb{R}^{B \times D \times H \times W}. \tag{1}$$

To emulate adversarial inter-person shifts, we inject two complementary perturbations at the spectral channel level. The first is variance-weighted modulation, emphasizing channels with unstable energy dispersion: $\mathbf{E}_{\text{var}} = \beta \cdot \text{Var}_{h,w}(\mathbf{E}) \odot \text{sign}(\mathbf{E})$. Then, the second perturbation introduces stochastic perturbations scaled adaptively by spectral variance statistics: $\mathbf{E}_{\text{pert}} = \alpha \cdot \epsilon \odot \eta$, $\epsilon_d = \text{clamp}\big(\text{Var}_{h,w}(\mathbf{E}_d), \underline{\epsilon}, \overline{\epsilon}\big)$, where $\eta \sim \mathcal{N}(0, I)$, and $\mathbf{E}_d \in \mathbb{R}^{B \times H \times W}$ is the $d$-th spectral slice. Totally, the perturbed spectrum is reconstructed as: $\mathbf{E}_{\text{rob}} = \tanh\Big(\mathbf{E} + \mathbf{E}_{\text{var}} + \mathbf{E}_{\text{pert}}\Big)$, where an adaptive Tanh (Zhu et al., 2025) enforces bounded activations and mitigates spectral collapse. To further refine invariance, a lightweight convolutional operator produces the attention map as: $\mathcal{A}_s = \sigma\big(\text{Conv2D}(\mathbf{E}_{\text{rob}})\big)$, which is broadcast along the temporal axis and injected back to the original tensor through residual masking and can be represented as:

$$x_s = x \cdot \text{Broadcast}_T^C(\mathcal{A}_s) + x. \tag{2}$$

Thus, the frequency branch constructs a distributionally robust spectral representation that resists person-specific distortions and preserves person-independent invariance.

### 3.1.2 TEMPORAL BRANCH: CONSISTENT DYNAMICS VIA REGULARIZED TRANSPORT

In *TFAM*, the temporal pathway focuses on enforcing consistency in dynamic evolution across heterogeneous individuals, aiming to suppress frame-level jitter, rhythm perturbations, and person-dependent irregularities. To achieve this, we construct a regularized transport kernel that jointly models global and local temporal correlations while explicitly controlling distributional sharpness and imbalance. In particular, given the input sequence $x \in \mathbb{R}^{B \times T \times C \times H \times W}$, two temporal affinity maps are derived via average- and max-pooling along spatial channels, denoted as: $\mathbf{M}_{\text{avg}}, \mathbf{M}_{\text{max}} \in \mathbb{R}^{B \times T \times T}$. After mean-centering each map, an adaptive temperature $\tau$ is estimated from their joint variance, which can be denoted as:

$$\tau = \tau_{\min} + \tau_{\text{scale}} \cdot \sqrt{\tfrac{1}{2}\big(\text{Var}(\mathbf{M}_{\text{avg}}) + \text{Var}(\mathbf{M}_{\text{max}})\big) + \varepsilon}, \tag{3}$$

where higher variance enlarges $\tau$, producing smoother transition probabilities under unstable temporal dynamics. The transport matrices are then constructed as: $\mathbf{K}_g = \text{Softmax}\Big(-\frac{(\bar{\mathbf{M}}_{\text{avg}})^2}{\tau}\Big)$, $\mathbf{K}_l = \text{Softmax}\Big(-\frac{(\bar{\mathbf{M}}_{\text{max}})^2}{\tau}\Big)$, and further symmetrized to preserve temporal reciprocity: $\mathbf{K}_g \leftarrow \tfrac{1}{2}(\mathbf{K}_g + \mathbf{K}_g^\top)$, $\mathbf{K}_l \leftarrow \tfrac{1}{2}(\mathbf{K}_l + \mathbf{K}_l^\top)$. By using the Sinkhorn–Knopp (Oquab et al., 2024) algorithm to approximate doubly-stochastic kernels, both matrices are subsequently normalized to ensure balanced transport across frames. To integrate local and global correlations, adaptive weights $\lambda_g, \lambda_l$ are predicted from row–column variances, yielding: $\mathbf{K}_{\text{mix}} = \lambda_g \mathbf{K}_g + \lambda_l \mathbf{K}_l$. To avoid degenerate peaky alignments that overfit dominant transitions, a Wasserstein-style (Arjovsky et al., 2017) deviation penalty is applied: $\mathbf{K}_{\text{mix}} \leftarrow \mathbf{K}_{\text{mix}} - \gamma_W \cdot \Big(\mathbf{K}_{\text{mix}} - \text{mean}_j(\mathbf{K}_{\text{mix}})\Big)_1$, where $(\cdot)_1$ denotes the absolute deviation averaged over index $j$. The regularized kernel is finally transformed into an attention map: $\mathcal{A}_t = \sigma(\Gamma \mathbf{K}_{\text{mix}} + \beta)$, which undergoes a linear projection and refinement through a self-attention block to capture higher-order temporal dependencies. The attended features are injected back into the sequence by residual broadcasting:

$$x_t = x \cdot \text{Broadcast}_C^{H,W}\big(\text{SelfAttn}(\text{Linear}(\mathcal{A}_t))\big) + x. \tag{4}$$

Totally, the temporal branch establishes stable and invariant temporal interactions, effectively aligning dynamics across diverse individuals under distributional uncertainty.

Table 1: Quantitative comparison on Micro-Action 52. **Bold**: Best, Underline: Second best.

| Method | | Body Top-1 | Action Top-1 | Top-5 | Body F1$_{macro}$ | F1$_{micro}$ | Action F1$_{macro}$ | F1$_{micro}$ | All F1$_{mean}$ |
|---|---|---|---|---|---|---|---|---|---|
| TIN (Shao et al., 2020) | AAAI | 73.26 | 52.81 | 85.37 | 66.99 | 73.26 | 39.82 | 52.81 | 58.22 |
| TimesFormer (Bertasius et al., 2021) | ICML | 69.17 | 40.67 | 82.67 | 61.90 | 69.17 | 34.38 | 40.67 | 51.53 |
| Video Swin T (Liu et al., 2021b) | CVPR | 77.95 | 57.23 | 87.99 | 71.25 | 77.95 | 38.53 | 57.23 | 61.24 |
| AAGCN (Shi et al., 2020) | T-IP | 74.13 | 56.96 | 84.37 | 65.88 | 74.13 | 41.36 | 56.96 | 59.58 |
| MS-G3D (Liu et al., 2020) | CVPR | 71.21 | 52.70 | 82.33 | 63.16 | 71.21 | 38.78 | 52.70 | 56.46 |
| CTR-GCN (Chen et al., 2021b) | ICCV | 76.01 | 59.06 | 86.05 | 68.46 | 76.01 | 43.38 | 59.06 | 61.73 |
| ST-GCN++ (Duan et al., 2022) | ACM MM | 72.04 | 53.78 | 82.04 | 62.95 | 72.04 | 37.52 | 53.78 | 56.57 |
| HD-GCN (Lee et al., 2022) | ICCV | 75.76 | 60.19 | 86.90 | 67.32 | 75.76 | 44.50 | 60.19 | 61.94 |
| Koopman (Wang et al., 2023) | CVPR | 75.04 | 59.70 | 86.79 | 66.48 | 75.04 | 44.57 | 59.70 | 61.45 |
| FR-Head (Zhou et al., 2023) | CVPR | 76.35 | 61.17 | 86.99 | 68.88 | 76.35 | 47.43 | 61.17 | 63.46 |
| SkateFormer (Do & Kim, 2025) | ECCV | 75.67 | 59.76 | 87.27 | 68.33 | 75.67 | 45.58 | 59.76 | 62.34 |
| Uniformer (Li et al., 2023) | T-PAMI | 79.03 | 58.89 | 87.29 | 71.80 | 79.03 | 48.01 | 58.89 | 64.43 |
| MANet (Guo et al., 2024) | T-CSVT | 78.95 | 61.33 | 88.83 | 72.87 | 78.95 | 49.22 | 61.33 | 65.59 |
| MMN (Gu et al., 2025b) | ACM MM | 78.52 | 62.71 | 89.83 | 71.86 | 78.52 | 48.27 | 62.71 | 65.34 |
| *PIUmr* (Ours) | | **80.95** | **63.18** | **89.87** | **75.51** | **80.95** | **52.79** | **63.18** | **68.11** |

### 3.1.3 CONSISTENCY-DRIVEN ALIGNMENT FOR PERSON INDEPENDENCE

To avoid representation drift and effectively enhance person-independent robustness, we introduce a perturbation-stability alignment scheme that adaptively weights temporal and frequency branches. The key idea is to evaluate each branch's invariance under characteristic perturbations and assign higher importance to the more stable one. For the temporal pathway, we generate a perturbed sequence $\hat{x}_t$ via random masking or shuffling, which can be expressed as: $s_t = \cos\left(\text{mean}(\phi_t(x)), \text{mean}(\phi_t(\hat{x}_t))\right)$, where $\phi_t(\cdot)$ is the temporal encoder. For the frequency pathway, variance-scaled Gaussian noise is injected into spectral channels: $\hat{x}_s = \psi_s(x) + \sum_{d=1}^{D} \epsilon_d \cdot \eta_d$, where $\eta_d \sim \mathcal{N}(0, I)$, and the corresponding stability is measured as: $s_s = \cos\left(\text{mean}(\psi_s(x)), \text{mean}(\hat{x}_s)\right)$. Then, the Normalized scores are designed to yield adaptive fusion weights as follows: $\lambda_t = \frac{s_t}{s_t + s_s + \varepsilon}$, and $\lambda_s = \frac{s_s}{s_t + s_s + \varepsilon}$, and the final representation is:

$$x_{\text{out}} = \lambda_t \cdot x_t + \lambda_s \cdot x_s. \tag{5}$$

This consistency-driven alignment privileges the more perturbation-resilient branch, mitigating asymmetric drift and ensuring robust integration across individuals.

### 3.2 GROUP-INVARIANT REGULARIZED LOSS FOR ROBUST DISCRIMINATION

To further promote person-independent robustness to a high level, we introduce *GIRL*, which interprets each mini-batch as a stochastic mixture of latent person-specific subgroups and imposes an optimization criterion that jointly emphasizes boundary instances and equalizes subgroup risks. Specifically, let $\{(\mathbf{f}_i, y_i)\}_{i=1}^{B}$ denote the mini-batch, where $\mathbf{f}_i \in \mathbb{R}^C$ are $\ell_2$-normalized pre-classifier features and $y_i$ the ground-truth labels. Then, the pairwise similarity is defined as $s_{ij} = \frac{1}{\tau} \mathbf{f}_i^\top \mathbf{f}_j$, with the indicator $\mathbb{I}_{ij} = \mathbb{K}[y_i = y_j]$ specifying positive relations. To effectively approximate hidden heterogeneity, indices are randomly permuted and partitioned into $G$ pseudo-groups $\{\mathcal{I}_g\}_{g=1}^{G}$ of size $\approx B/G$, where all subsequent computations are restricted within each subgroup, thereby truly simulating person-dependent distributional subsets.

Within a group $\mathcal{I}_g$, to accentuate moderately hard positives and suppress trivial ones, we design a Gaussian-shaped reweighting (Wu et al., 2023) on pairwise similarities. Specifically, for the anchor $i \in \mathcal{I}_g$, $w_{ij}^{(g)} = \frac{\exp\left(-\frac{1}{2}\left(\frac{s_{ij} - \eta}{\eta}\right)^2\right)}{\sum_{k \in \mathcal{I}_g} \exp\left(-\frac{1}{2}\left(\frac{s_{ik} - \eta}{\eta}\right)^2\right)}$, $\eta$ specifies the adversarial neighborhood around where the emphasis is applied. The weighted group objective can then be represented as:

$$\mathcal{L}_g = \frac{1}{|\mathcal{I}_g|} \sum_{i \in \mathcal{I}_g} \frac{-\sum_{j \in \mathcal{I}_g} \mathbb{I}_{ij} w_{ij}^{(g)}\left(s_{ij} - \log \sum_{k \in \mathcal{I}_g} e^{s_{ik}}\right)}{\sum_{j \in \mathcal{I}_g} \mathbb{I}_{ij} + \varepsilon}. \tag{6}$$

Table 2: Ablation study of *TFAM* and *GIRL* in the proposed *PIUmr* on MA-52

| Setting | PIUmr | | Body | | Action | | All |
| | TFAM | GIRL | $F1_{micro}$ | $F1_{macro}$ | $F1_{micro}$ | $F1_{macro}$ | $F1_{mean}$ |
|---|---|---|---|---|---|---|---|
| a | ✗ | ✗ | 73.74 | 79.99 | 52.39 | 62.30 | 67.11 |
| b | ✔ | ✗ | 74.79 | 80.45 | 51.80 | 62.85 | 67.47 |
| c | ✗ | ✔ | 74.67 | 80.76 | 52.46 | 62.87 | 67.69 |
| d | ✔ | ✔ | **75.51** | **80.95** | **52.79** | **63.18** | **68.11** |

Next, the group-contrastive risk is aggregated over all groups as: $\mathcal{L}_{\text{grp}} = \frac{1}{G} \sum_{g=1}^{G} \mathcal{L}_g$. To prevent collapse onto dominant subgroups and to balance learning difficulty across heterogeneous subsets, we introduce a group-invariant regularizer that penalizes the dispersion of group-wise risks. Let $r_g = \text{stopgrad}(\mathcal{L}_g)$ be the detached risk of group $g$, the variance penalty is then defined as: $\mathcal{R}_{\text{var}} = \text{Var}\big(\{r_g\}_{g=1}^{G}\big)$. The final *GIRL* objective thus becomes:

$$\mathcal{L}_{\text{GIRL}} = \mathcal{L}_{\text{grp}} + \lambda_{\text{var}} \, \mathcal{R}_{\text{var}}. \tag{7}$$

This formulation effectively steers optimization toward ambiguous and boundary samples that dominate generalization errors, while the variance penalty enforces equilibrium across pseudo-groups as a proxy for worst-case regularization. Therefore, *GIRL* achieves subgroup-invariant risk minimization, enhancing person-independent discrimination without relying on explicit subject identities.

## 4 EXPERIMENTS

### 4.1 EXPERIMENTAL SETUP

**Dataset.** We evaluate the system performance on the large-scale Micro-Action 52 (MA-52) dataset (Guo et al., 2024), which contains 22422 samples across 52 fine-grained action categories and 7 body-level categories. Captured from professional interview scenarios, MA-52 emphasizes subtle whole-body dynamics and provides a comprehensive benchmark for distributional robustness in micro-action recognition. For a fair comparison, we follow the official split with 11250 training, 5586 validation, and 5586 test instances.

**Evaluation Metrics.** To comprehensively demonstrate the performance, we report Top-1/Top-5 accuracy and F1 score. For F1, both macro (unweighted average across classes) and micro (sample-level average) are computed at body- and action-levels. To provide a unified measurement, we define $F1_{\text{mean}} = \frac{F1_{macro}^{body} + F1_{micro}^{body} + F1_{macro}^{action} + F1_{micro}^{action}}{4}$, which jointly evaluates recognition across both granularities.

**Implementation Details.** All models are trained on NVIDIA RTX A6000 GPUs using PyTorch with mixed precision. Each video clip contains 16 frames, resized to $224 \times 224$. We use the AdamW optimizer with an initial learning rate of $1 \times 10^{-3}$, weight decay of $1 \times 10^{-4}$, batch size of 80, and train for 120 epochs. The learning rate is reduced by a factor of 0.1 at the 30th and 60th epochs. The training objective is the sum of cross-entropy loss and *GIRL*.

### 4.2 COMPARISON WITH THE STATE-OF-THE-ART METHODS

As shown in Tab. 1, although existing 3D CNNs, Transformers, and GCN-based approaches achieve reasonable results, they inevitably suffer clear degradation when transferring from body-level to action-level evaluation. Since body-level emphasizes cross-person recognition and action-level highlights fine-grained category discrimination, this gap indicates that conventional architectures cannot simultaneously generalize across individuals and handle subtle intra-class variations. In contrast, our proposed *PIUmr* delivers consistent improvements on both levels. The gains at the body-level demonstrate that disentangling temporal and frequency signals with alignment under perturbations effectively reduces inter-person variability, leading to stronger person-independent generalization. At the same time, the advantages at the action-level confirm the benefit of subgroup-invariant risk regularization, which enhances robustness on boundary and uncertain cases. Overall, *PIUmr* improves recognition accuracy while achieving a more balanced robustness across person-independent and fine-grained evaluations, validating its capacity to mitigate representation shift beyond what backbone scaling alone can achieve.

Table 3: Ablation study of frequency, temporal branches, and fusion mechanism in *TFAM*.

| Setting | TFAM | | | Body | | Action | | All |
|---|---|---|---|---|---|---|---|---|
| | Fre. | Tim. | Fus. | $F1_{micro}$ | $F1_{macro}$ | $F1_{micro}$ | $F1_{macro}$ | $F1_{mean}$ |
| a | ✗ | ✗ | ✗ | 74.67 | 80.76 | 52.46 | 62.87 | 67.69 |
| b | ✔ | ✗ | ✗ | 74.79 | 80.65 | 52.54 | 62.84 | 67.71 |
| c | ✗ | ✔ | ✗ | 74.58 | 80.43 | 52.60 | 62.41 | 67.51 |
| d | ✔ | ✔ | ✗ | 75.09 | 80.84 | 52.79 | 62.94 | 67.92 |
| e | ✔ | ✔ | ✔ | **75.51** | **80.95** | **52.79** | **63.18** | **68.11** |

## 4.3 ABLATION STUDIES

**Effectiveness of Each Module.** As shown in Tab. 2, removing either module causes clear performance degradation, especially at the action level where subtle distinctions are most sensitive to representation shift. Introducing *TFAM* alone improves body-level results by aligning temporal and frequency signals, effectively reducing inter-person variability and enhancing person-independent generalization. However, without explicit risk balancing, decision boundaries for ambiguous samples remain fragile. Conversely, applying only *GIRL* improves action-level recognition by enforcing subgroup-invariant optimization, but the absence of robust temporal–frequency disentanglement leaves subject-specific perturbations unresolved. The best results arise when both modules are integrated in *PIUmr*, where *TFAM* stabilizes representation learning against person-dependent noise and *GIRL* balances risks across latent subgroups, yielding consistent gains across body- and action-level metrics and the most robust performance.

**Effectiveness of Temporal–Frequency Alignment in *TFAM*.** As shown in Tab. 3, relying only on the temporal branch reduces rhythm jitter but fails to address spectral sensitivity, while focusing only on the frequency branch mitigates structural drift but overlooks dynamic inconsistencies. Simply combining the two improves robustness, yet naive fusion cannot adapt to asymmetric perturbations across modalities. With the proposed consistency-driven alignment, temporal dynamics and spectral features are adaptively balanced, harmonizing stability across individuals and suppressing residual cross-modal drift. This design achieves the most reliable gains at both body- and action-level evaluations, confirming that temporal modeling, frequency modeling, and consistency-based fusion are mutually complementary. Their joint integration is essential for person-independent and fine-grained robustness in MAR.

## 4.4 VISUALIZATION.

### 4.4.1 T-SNE VISUALIZATION

We present t-SNE (van der Maaten & Hinton, 2008) plots of the learned embeddings in Fig. 4. Compared with the baseline, where categories overlap heavily, our *PIUmr* yields more distinct and clearly separated clusters, reflecting stronger inter-class discrimination and robustness under subject variability. Notably, some coarse categories under *PIUmr* exhibit multi-centric patterns rather than forming a single compact cluster. This arises because visualization is based on coarse-grained annotations, whereas training uses

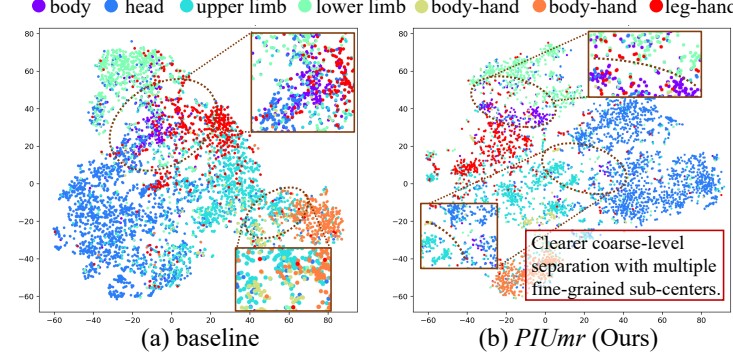

(a) baseline     (b) *PIUmr* (Ours)

Figure 4: Illustration of feature distribution learned by the baseline and our *PIUmr* on MA-52.

fine-grained labels, naturally producing sub-cluster structures. In addition, the subgroup modeling in *GIRL* partitions samples into pseudo-groups to approximate hidden person-specific distributions, further encouraging multi-center formations. Importantly, these sub-clusters remain tightly bounded within their respective coarse categories, showing that intra-class cohesion is preserved despite finer structural divisions. These results confirm that *PIUmr* not only enlarges inter-class margins but also

maintains stable intra-class compactness, thereby supporting fine-grained recognition and person-independent generalization simultaneously.

### 4.5 ANALYSIS OF SIMILARITY DISTRIBUTIONS

To further analyze the representational behavior of our framework, we compare the cosine similarity distributions of inter- and intra-class pairs in Fig. 5. The baseline shows a right-shifted inter-class distribution, indicating excessive similarity across categories and blurred decision boundaries under subject variability. In contrast, *PIUmr* produces a left-shifted inter-class distribution with enlarged margins, suggesting stronger separability even when individuals exhibit diverse motion styles. On the intra-class side, baseline features present dispersed similarity values, reflecting weak cohesion and sensitivity to person-specific noise. Our framework instead yields a sharper, more concentrated distribution, demonstrating improved intra-class compactness and robustness against perturbations. These findings confirm that by combining temporal–frequency alignment in *TFAM* with subgroup-invariant regularization in *GIRL*, *PIUmr* not only

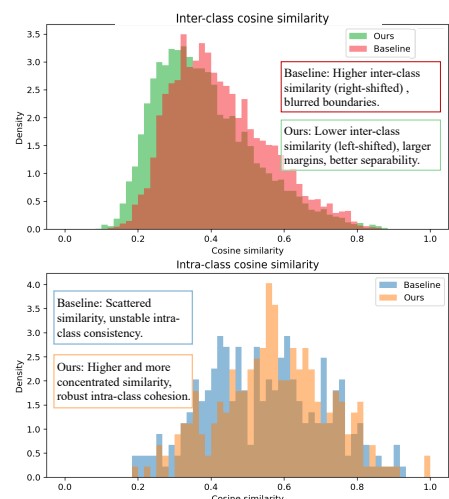

Figure 5: Visualization of inter- and intra-class cosine similarity distributions on MA-52.

enlarges inter-class margins but also stabilizes intra-class clustering, thereby enabling fine-grained discrimination and reliable person-independent generalization.

### 4.6 VISUALIZATION OF LOSS LANDSCAPE

We further compare the loss landscapes (Li et al., 2024) of the baseline and *PIUmr* on MA-52 in Fig. 6. The baseline exhibits a rugged and sharply curved surface with irregular local minima, reflecting unstable optimization and vulnerability to subject-specific perturbations. In contrast, *PIUmr* yields a smoother and flatter landscape with wider basins, suggesting more stable convergence and improved

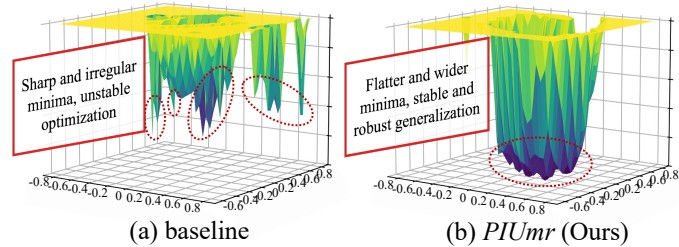

(a) baseline     (b) *PIUmr* (Ours)

Figure 6: The visualization of loss landscape.

generalization across heterogeneous individuals. This evidence confirms that explicitly mitigating representation shift through temporal–frequency alignment and subgroup-invariant regularization regularizes the learning dynamics, thereby enhancing person-independent robustness in MAR.

## 5 CONCLUSION

We propose *PIUmr*, a person-independence universal framework for MAR that explicitly addresses representation instability arising from inter-subject variability. The framework incorporates two plug-and-play modules, *i.e.*, the *TFAM*, which disentangles temporal dynamics and frequency structures and adaptively aligns them via consistency-driven fusion, thereby producing more stable representations under heterogeneous perturbations; and the *GIRL*, which enforces subgroup-invariant regularization by reweighting hidden subsets and constraining variance across pseudo-groups, thus enhancing robustness on boundary and uncertain cases. Through organic combination, these components form a unified DRO-inspired paradigm that improves both stability and discriminability in fine-grained, cross-subject recognition. Extensive experiments on MA-52 confirm the effectiveness of *PIUmr*, yielding state-of-the-art performance with smoother optimization behavior. In future work, we plan to extend this framework to broader multi-modal behavioral analysis and explore lightweight variants for real-world deployment.

ETHICS STATEMENT

This work strictly follows the ICLR Code of Ethics. All experiments are conducted on the publicly available MA-52 dataset, which was released for academic research purposes. No new data collection or human subject experiments were performed. The dataset contains no personally identifiable information, and our study involves no privacy or security risks. The proposed framework aims to improve robustness and person-independence in micro-action recognition, with potential applications in areas such as psychological assessment and human–computer interaction. We confirm that the research complies with standards of research integrity and responsible data usage, and does not pose harmful or discriminatory implications.

REPRODUCIBILITY STATEMENT

We have made every effort to ensure the reproducibility of this work. The architecture of the model, as well as the implementation details and mathematical formulations of the TFAM and GIRL modules, are described in detail in the methodology section. The experimental setup systematically specifies the use of the publicly available MA-52 dataset, including the dataset splits, evaluation metrics, training settings, and preprocessing steps, all of which are clearly documented in the main text. To further facilitate reproducibility, our source code and training scripts will be released upon acceptance of the paper. With these resources, researchers will be able to reproduce the main results and findings of this work under the same dataset and experimental settings.

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

## USE OF LARGE LANGUAGE MODELS (LLMS)

We used Large Language Models (LLMs) solely as an auxiliary tool to aid and polish the writing of this manuscript. The models were not involved in research ideation, experimental design, data analysis, or result interpretation. Their role was limited to improving clarity, grammar, and readability of the text.

