# OpenReview forum: "Every Subtlety Counts: Fine-grained Person Independence Micro-Action Recognition via Distributionally Robust Optimization"
_ICLR.cc/2026/Conference — ICLR 2026 Conference Withdrawn Submission_

### Official Review · Reviewer_C8Go · 2025-10-27

**Soundness:** 2
**Presentation:** 2
**Contribution:** 2
**Rating:** 2
**Confidence:** 3

**Summary:**

The paper proposes PIUmr (TFAM + GIRL) to improve person-independent micro-action recognition via temporal/frequency alignment and a pseudo-group DRO-style regularizer. Experiments on MA-52 show improvements and a range of visualizations, but empirical and methodological gaps limit confidence in the claims.

**Strengths:**

- Well motivated problem (person-independence in micro-actions).
- Modular design (TFAM fusion and GIRL regularizer) with extensive visual diagnostics.
- Empirical gains.

**Weaknesses:**

1. **Novelty is limited** — DRO has been applied to domain generalization in vision (e.g., recent works on fine-grained recognition), and person-independence in action recognition overlaps with existing surveys on HAR variability.
2. **Limited baseline comparisons (OOD/DRO)** — no direct comparison to standard OOD/group-robust methods (e.g., GroupDRO, Sagawa, CORAL/Fish variants).
3. **Single dataset / limited generalization** — all results are on MA-52; no cross-dataset transfer or comprehensive leave-one-subject-out experiments to support general person-independence claims.
4. **No variance / statistical significance** — tables report single numbers only. Report mean ± std over multiple seeds and, where relevant, statistical tests.
5. **Sensitivity to model size and robustness** — no experiments showing how the method scales or how robust the gains are across small → large backbones.
6. **Hyperparameter & pseudo-group sensitivity under-analyzed** — GIRL depends on many choices. Paper lacks a default hyperparameter table and sensitivity sweeps; results may be brittle.
7. **Missing worst-case / subgroup metrics** — method is framed as DRO-inspired but only global averages are shown. Need per-subject / per-group worst-case numbers and minority-class performance to validate DRO claims.
8. **Theoretical clarity lacking** — the DRO connection is mainly heuristic. Either provide a short formal/intuition proposition linking GIRL to worst-case group risk or tone down DRO claims.
9. **Compute / memory / latency costs omitted** — TFAM uses Sinkhorn, attention, DCT banks and GIRL uses pairwise group ops. Report GFLOPs, training time/epoch, GPU memory, and inference overhead.
10. **Reproducibility & algorithmic clarity** — method description is fragmented. Add an algorithm box (pseudocode), a hyperparameter table, and exact training recipes (seeds, epochs, early-stop, optim settings).

**Questions:**

See above.

---

### Official Review · Reviewer_wZiQ · 2025-10-27

**Soundness:** 4
**Presentation:** 3
**Contribution:** 3
**Rating:** 8
**Confidence:** 4

**Summary:**

This paper presents PIUmr, a distributionally robust framework for fine-grained micro-action recognition, addressing cross-person variability in motion style, rhythm, and structure. The framework combines temporal–frequency alignment with a group-invariant regularization loss, improving generalization under distribution shifts. Experiments on the large-scale MA-52 dataset demonstrate consistent gains over baselines. Additionally, analyses on hyperparameter sensitivity and cross-dataset robustness further strengthen the claims.

**Strengths:**

1. The paper proposes a well-structured and balanced framework for person independent micro-action recognition. Its integration of TFAM and GIRL is conceptually sound and empirically effective, achieving strong robustness and generalization with clear experimental support.

2. The paper introduces PIUmr, a distributionally robust framework for fine-grained micro-action recognition, addressing cross-person variability in motion style and structure.

3. The integration of a temporal–frequency alignment module and a group-invariant regularization loss is conceptually clear and theoretically grounded, leading to consistent improvements under distribution shifts.

4. Experiments on the MA-52 dataset are comprehensive, and the visual analyses (t-SNE, similarity maps, and loss landscapes) convincingly demonstrate improved feature consistency and optimization stability.

5. The proposed framework demonstrates a well-balanced design between performance and robustness; the modular formulation of TFAM and GIRL not only improves cross-person stability but also shows potential extensibility to other temporal recognition tasks.

**Weaknesses:**

1. In terms of theoretical details, the explanation of the subgroup variance regularization in the GIRL module is somewhat brief; providing additional discussion on its convergence behavior during training would strengthen the theoretical soundness.

2. Moreover, the sensitivity analysis of key hyperparameters, such as the number of pseudo-groups G, has not been reported, which may raise concerns regarding reproducibility.

3. The description of Figure 5 is somewhat ambiguou. It remains unclear whether the vertical axis of the similarity curves represents relative distance or normalized similarity. Clarifying this definition in the caption would help readers interpret the model’s feature discriminability more accurately.

4. In the GIRL loss formulation (which includes the subgroup risk term and the variance regularization term), the meanings of some symbols, for example, the correspondence between subgroup indices and the overall batch risk, are not fully explained.

5. Some handwriting typos and grammatical errors need to be carefully checked.

**Questions:**

Please see Weaknesses.

---

### Official Review · Reviewer_QQpD · 2025-10-28

**Soundness:** 2
**Presentation:** 2
**Contribution:** 2
**Rating:** 2
**Confidence:** 4

**Summary:**

This paper introduces the Person Independence Universal Micro-action Recognition Framework (PIUmr). The PIUmr has two synergistic  components: The Temporal–Frequency Alignment Module (TFAM) at the feature level, which normalizes person-specific motion characteristics through a dual-branch architecture, mitigating variations in motion styles and physical attributes. The Group-Invariant Regularized Loss (GIRL) at the loss level, which guides robust learning.

**Strengths:**

This paper introduces the principle of Distributionally Robust Optimization into Micro-action recognition explicitly from the perspective of person independence.

The designed module is plug-and-play.

**Weaknesses:**

Experiemnts is limited on the MA-52 dataset, which makes me doubt the effectiveness of this method. If the proposed module is pug-and-play, more experiments should be conducted to verify its effectivess and generalization.

Text in Figure 1, 2, 3 are too small and hard to identify.

The designed modules, which operate at both the feature level and the loss level, are common examples of an A + B structure. Therefore, the proposed method is meaningless.

The addressed problem is very limited. Micro-action Recognition is an old and well-studied task.

The related works are poorly summarized, e.g.,

2.1 MICRO-ACTION RECOGNITION AND TEMPORAL–FREQUENCY MODELING. This topic is too long, and what is the focus? I suggest the authors carefully reviewed the recent papers of  MICRO-ACTION RECOGNITION.

2.2 DISTRIBUTIONALLY ROBUST OPTIMIZATION IN VISION. This topic is too broad, and cannot be reviewed with a single paragraph.

**Questions:**

It is recommended that the authors choose more meaningful research problems to investigate or delve deeper into identifying unresolved critical issues in Micro-action Recognition. The current method offers no meaningful contribution to the field and merely represents a typical "A + B" approach without substantive innovation.

---

### Official Review · Reviewer_Qy7F · 2025-10-31

**Soundness:** 3
**Presentation:** 3
**Contribution:** 2
**Rating:** 2
**Confidence:** 5

**Summary:**

This paper addresses the problem of person-independent micro-action recognition, where subtle and short-duration human motions vary significantly across individuals. The authors model inter-subject variation as a form of distributional shift and propose a framework called PIUmr that combines two key modules: 1) TFAM (Temporal–Frequency Alignment Module) for stabilizing temporal and spectral motion features across subjects. 2) GIRL (Group-Invariant Regularized Loss) to enforce distributionally robust optimization via pseudo-group regularization. Experiments are conducted on one public micro-action dataset, showing improved cross-subject recognition compared with several baseline methods.

**Strengths:**

+ The paper clearly presented the motivation for improving cross-person generalization in MAR tasks, which is an important and underexplored challenge.
+ The formulation of person variability as a distributional robustness problem is conceptually appealing and connects MAR with ideas from DRO and domain generalization.
+ The paper is well written and structured, with clear diagrams explaining the overall framework and loss formulation.

**Weaknesses:**

- The technical contributions, including feature-alignment module and group-invariant loss, represent a minor architectural extension of existing robust/domain-generalized HAR or MAR frameworks. Similar designs appear in prior works such as domain-adversarial training and group DRO. The novelty lies mainly in combining known techniques rather than introducing a fundamentally new idea.
- The authors evaluated the proposed method only on a single benchmark dataset, MA-52. Given the claim of generalizable, person-independent robustness, evaluation on multiple MAR or HAR benchmarks would be essential to demonstrate robustness and significance.
- Given the narrow experimental scope and incremental method, the overall contribution may fall short of ICLR’s standard for conceptual or empirical novelty.

**Questions:**

1. Could the authors clarify how the proposed DRO formulation differs in practice from existing GroupDRO or domain-adversarial training objectives?
2. Have the authors tested the method on any other micro-action datasets or synthetic variations to validate generalization beyond MA-52?
3. How sensitive is the model to group definition in the GIRL loss—does it rely on true subject IDs, or can it handle unsupervised grouping?

---

### Note · Authors · 2025-11-12

I have read and agree with the venue's withdrawal policy on behalf of myself and my co-authors.